# SynthCraft: An AI partner for synthetic data generation to support data access and augmentation in healthcare

Thomas Callender[1, 2*], Anders Boyd[3,4,5], Robert Davis[2,6], Silas Ruhrberg Estevez[7], Juan M. Lavista Ferres[8], Mihaela van der Schaar[2,6*]

**1** Department of Public Health and Primary Care, University of Cambridge, Cambridge, United Kingdom, **2** Cambridge Centre for AI in Medicine, University of Cambridge, Cambridge, United Kingdom, **3** Department of Infectious Diseases, Inselspital, Bern University Hospital, University of Bern, Bern, Switzerland, **4** Amsterdam UMC location University of Amsterdam, Department of Infectious Diseases, Meibergdreef 9, Amsterdam, The Netherlands, **5** Amsterdam Institute for Immunology and Infectious Diseases, Infectious Diseases, Amsterdam, The Netherlands, **6** Department of Applied Mathematics and Theoretical Physics, University of Cambridge, Cambridge, United Kingdom, **7** School of Clinical Medicine, University of Cambridge, Cambridge, United Kingdom, **8** AI for Good Lab, Microsoft, Redmond, Washington, United States of America

* tac68@cam.ac.uk (TC), mv472@damtp.cam.ac.uk (MvdS)

## Abstract

Access to high-quality data provides the foundation for biomedical research. But data access is often limited or challenging due to privacy constraints, whilst the data themselves may be unrepresentative or sparse. Synthetic data can support both privacy-preserving data access and advanced analytical workflows, including data augmentation or the development of digital twins. However, the use of synthetic data remains limited due to the complexity of the methods themselves, their use, and their evaluation. To address this, we developed SynthCraft, an AI tool to support the principled, transparent, application of state-of-the-art synthetic data generation methods. SynthCraft couples a reinforcement learning-based reasoning engine with large language models (LLMs) to orchestrate the workflow necessary for the generation of synthetic data based on dynamic interaction with the user through natural language. We demonstrate the capability of SynthCraft with both tabular and genomic datasets: the National Health and Nutrition Examination Survey (NHANES) and the Cancer Genome Atlas (TCGA). Using SynthCraft, we analysed the privacy, statistical fidelity, and downstream utility of four different synthetic data generators both with and without explicit privacy-preserving designs when applied to both the NHANES and TCGA datasets. We show that how different generators perform differently – and that no single method was optimal – across varying use-cases and datasets. Furthermore, we demonstrate how SynthCraft can be used for data augmentation as part of a workflow to attempt to mitigate imbalances in the proportion of individuals from different ethnic backgrounds. In conclusion, a human-in-the-loop AI partner using LLMs can support the generation of synthetic datasets. Such tools could improve the quality,

**Data availability statement:** Both the NHANES and TCGA datasets are publicly available from the following links: NHANES - https://wwwn.cdc.gov/nchs/nhanes/continuousnhanes/default.aspx?Cycle=2021-2023 – and TCGA – https://xenabrowser.net/datapages/?hub=https://tcga.xenahubs.net:443.

**Funding:** The author(s) received no specific funding for this work.

**Competing interests:** I have read the journal's policy and the authors of this manuscript have the following competing interests: A.B. reports receiving speaker's fees from Gilead Sciences, Inc. All other authors report no competing interests.

reproducibility, and transparency of research methods, whilst increasing their accessibility. Research into their use across different methodological areas is warranted.

## Author summary

Medical research depends on access to patient data, but legitimate privacy concerns often mean access is restricted. We created SynthCraft to address this challenge. SynthCraft is an AI partner designed to help researchers generate synthetic versions of medical datasets entirely through natural language, without requiring programming skills. Synthetic data mimic the patterns seen in real datasets, but without containing actual patient data. However, creating and evaluating synthetic data is technically complex, requiring specialised knowledge that limits its accessibility. SynthCraft supports users through each step in the generation of synthetic data: analysing the original data, selecting appropriate generation methods, creating synthetic data itself, before finally rigorously evaluating the results. All actions and code used by SynthCraft are recorded throughout. We demonstrated SynthCraft's capabilities using a national health survey and cancer genomics dataset. Models trained on our synthetic data performed comparably to those trained on real data. We also explored using synthetic data to address imbalances in ethnic representation, though we did not find that this improved model performance in these analyses. By making advanced methods accessible through natural language and ensuring transparent, reproducible workflows, such tools could transform how researchers apply state-of-the-art methods across biomedical research.

## Introduction

Healthcare research is premised on access to high-quality data. Nevertheless, legitimate privacy concerns mean healthcare data are often difficult to access if not entirely unavailable [1]. When available, data may be sparse, subject to biases, or unrepresentative [2]. The implications of this are felt throughout biomedical research.

Synthetic data has emerged as a powerful approach to overcome these problems. Rather than masking or anonymizing real data, synthetic data are generated to mirror the statistical patterns and relationships found in real datasets [2,3]. Because of this, synthetic data can support privacy-preserving data access and the development of digital twins [2–5]. When applied with care, synthetic data can also play a role in mitigating issues of fairness, biases, and data sparsity through data augmentation and adaptation [2,4]. While the benefits of synthetic data are increasingly recognised, its application in healthcare remains in its infancy [6]. Key challenges include the complexity of developing synthetic data, the speed with which synthetic data methodologies are being developed, and the lack of standardisation of the metrics by which synthetic data should be judged [7,8].

Efforts to improve the quality and accessibility of research methods have historically revolved around training, the development of software packages that abstract elements of how a method is implemented [9–11], and reporting guidelines [12]. Software for synthetic data have begun to bridge this gap [9–11], yet their use still demands advanced programming and familiarity with complex data pipelines; skills that many healthcare researchers do not possess or lack the time to apply effectively [13]. Guidelines have been used as both an educational tool and to improve the quality of research [12], but their impact has been mixed [14].

Here we introduce SynthCraft, a large language model (LLM)-based partner for synthetic data development as a solution to these barriers. This system uses LLMs to orchestrate multiple sequential or concurrent steps to solve complex problems. Working together entirely in natural language, SynthCraft empowers a researcher with state-of-the-art software libraries to build synthetic data by progressing through a principled, stepwise, approach, discussing problems and solutions as they arise. In developing this system, we first show the necessity of bespoke reasoning systems over using LLMs alone before comparing the performance of SynthCraft using different LLMs. We then demonstrate the capability of SynthCraft across tabular and genomic datasets for both data access and augmentation tasks.

## Methods

### Overview of SynthCraft

Designed to act as a "human-in-the-loop" partner, SynthCraft guides users through an interactive, step-by-step process (Fig 1). First, SynthCraft characterises the real dataset, identifying key features and the data structure, as well as performing exploratory data analyses. SynthCraft then engages the user to consider the most appropriate synthetic data generation methods for their circumstances, explaining the different strengths and weakness of alternative approaches, before

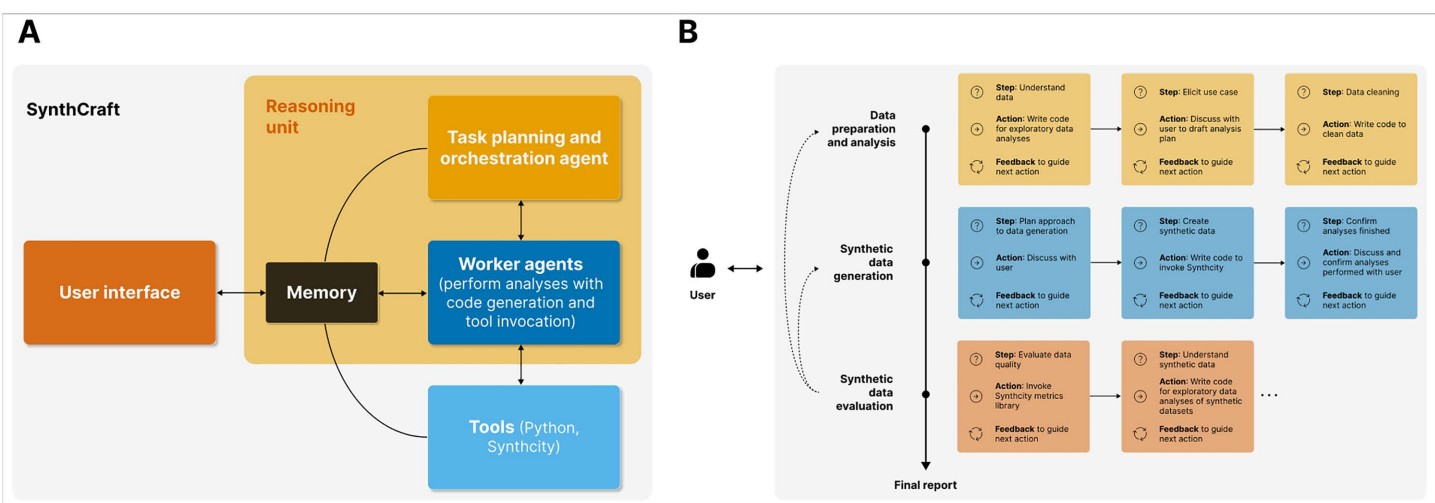

**Fig 1. Overview of SynthCraft. (A)** SynthCraft is a modular framework consisting of large language model (LLM) agents with access to tools (SynthCity, Python) linked with working memory and underpinned by a reasoning system for in-context learning. Interaction with the user is through a natural language interface. An illustration of SynthCraft's synthetic data generation pipeline is shown in (**B**) alongside a schematic of how the episodic multi-armed bandit reasoning approach works (to the right of the vertical line) [15,16]. At each stage of the pipeline, from data preparation and developing an analysis plan through evaluating the quality of the synthetic data generated, multiple intermediate steps may be needed. In the illustration, we have simplified for clarity, but at each step the agent has a particular state, reflecting progress towards completing the relevant task, can select appropriate actions, and then receives feedback either from interaction with the user, external tools invoked, or self-reflection [15,16]. The number of episodes – or steps – will vary depending on the use case, the dataset, and the interaction with the user. Though presented as a sequential pipeline, SynthCraft may need to return to earlier stages; for example, to trial alternative synthetic data generators if the quality of the synthetic data generated for a given task is insufficient at the evaluation stage. Subfigures adapted from refs [15,16].

invoking Synthcity to generate synthetic data itself [9]. Synthcity is an open-source package with extensive community engagement that provides a standardised interface for accessing and evaluating a comprehensive array of generators for any synthetic data use case – from synthetic data generation to managing privacy, fairness, domain adaptation and image generation [9] (Table A in S1 Text). Together, the user and partner compare the generated data and iteratively refine analyses based on user feedback. The process has been designed to minimise ambiguity, ensure methodological rigour, and quality-assure the generated synthetic data. User prompts and any code generated by SynthCraft can be saved directly whilst, on completion, a structured report detailing all steps taken, decisions made, and code run, is produced to support transparency and reproducibility.

SynthCraft is a modular LLM framework, in which a coordinating LLM agent controls worker agents with the ability to generate code and run specialised synthetic data generation tools. This framework includes several important guardrails: first, the LLMs are not responsible for writing code to generate or evaluate synthetic data, instead using Synthcity (version 0.2.12). Second, we protect against the introduction of errors at the point of generating synthetic data by ensuring the human user is asked to validate key parameters before Synthcity is invoked. More broadly, the user is kept in the loop throughout, can ask questions at any point, and can revert steps. We used GPT-5 from OpenAI (model: GPT-5; API version: "2025-08-07"; temperature: 0.5) on a secured instance of Azure as the underlying LLM in these analyses, but an alternative LLM could be simply interchanged to underpin the agents in this framework. The LLMs themselves are not privy to the real or synthetic data at any point, only prompts – the instructions provided by the user – whilst all analyses are undertaken on a user's own device.

The collaborative, sequential, decision-making process between the user and SynthCraft uses a reasoning system trained using multi-armed bandits, a type of reinforcement learning designed for complex multi-stage optimisation problems (Fig 1) [15,16]. Each stage in the process by which synthetic data are generated – which are encoded as fixed logic – can be considered as an episode consisting of one or more steps. The steps necessary for each episode will develop based on the needs of the user and characteristics of the data – i.e., adaptive learning [15]. For example, because the performance of synthetic data generation methods differs between use-cases and datasets, the generative step may require several iterative cycles before completion. At each step, an action is taken, such as running a synthetic data generator tool or producing a report. Each action is associated with both a cost and a reward. Should the task be completed or a problem encountered by the LLM that requires discussion with the user, a stop action occurs. Feedback received from all other actions taken within that episode informs the next action to be taken. The overall objective of the LLM is to maximise net rewards (total rewards less costs), which occurs on successful completion of the task. Further details are presented in Box 1.

## BOX 1: Details of the SynthCraft reasoning engine

SynthCraft integrates a reasoning engine that guides users through the process of synthetic data generation using a transparent, structured decision framework. Inspired by the CLiMB architecture [16], this reasoning process is formalised as an episodic multi-armed bandit [15], tailored to maximise utility (output quality) while minimising interaction cost (user burden).

The reasoning engine consists of several core components [15,16]:

- A set of tasks (**episodes**) necessary to complete an analysis plan.

- **Costs** associated with specific actions that are returned as **feedback.**

- **Actions** to complete the tasks, from invoking tools to asking the user for feedback.

- A **state** that corresponds to all previous actions within the episode along with their costs.

Let $E$ be the set of episode types or subtasks in the synthetic data pipeline, including:

1. Dataset intake and characterisation

2. Analytic intent elicitation

3. Synthetic data generation with or without specific privacy guarantees

4. Evaluation strategy selection

5. Iterative refinement

6. Transparent documentation

Note, data pre-processing is currently expected to be performed before data are passed to SynthCraft.

Each episode, $\rho$, corresponds to one subtask $e_\rho \in E$, and consists of a sequence of actions $(a_1^\rho, \ldots, a_t^\rho)$ drawn from a set of Actions, $\mathcal{A}$ [15,16]. These actions either progress the task or end the episode and engage the user. We enforce a maximum number of actions per episode before automatically discussing with the user to prevent infinite iteration [16].

The next action to be taken $a_t^\rho \in \mathcal{A}$ draws on the previous sequence of actions and the feedback received from each action [15]. Feedback is attributed a cost, $c$, of 0 or 1, and can be received from [16]:

- external tools (e.g., the results from statistical analyses; synthetic data generators);

- LLM self-reflection; or,

- the user themselves (either mid-episode or at the end of the episode).

Feedback that requires the user is penalised ($c = 1$) to minimise user burden. At the end of episode, there is a terminal reward $R^\rho$ again of 1 – corresponding to approval of the actions taken by SynthCraft – or 0 if the user requires the episode to be performed differently.

The reasoning agent maintains a dynamic plan over the subtasks. A subtask, $e_\rho$, is considered complete when the associated episode receives a reward $R^\rho = 1$. The plan is not strictly sequential; SynthCraft can reorder subtasks based on task context and user feedback, maintaining flexibility and robustness across different use cases. The ultimate objective of SynthCraft's reasoning system is to maximise terminal rewards by completing the analysis plan and its constituent steps efficiently with minimal user input [16]. All actions, feedback, and decisions are recorded to enable full reproducibility and auditability of the synthetic data workflow.

To analyse the added value of SynthCraft above using an LLM (GPT-5) in combination with Synthcity, we performed reasoning ablation studies. In each study, we replace the reasoning engine of SynthCraft with a coordinator agent that uses GPT-5. Here, the GPT-5 coordinating agent has access to the same tools, allowing us to analyse the impact of the SynthCraft beyond using LLMs on their own when running an end-to-end workflow to generate and evaluate synthetic data. Our assessments included whether relevant steps were performed and the quality of analyses, if errors were generated, as well as how or whether a human user is involved at critical stages in decision making.

## Datasets

To demonstrate the capacity of SynthCraft to generate synthetic datasets across both epidemiological data and genomic data, we used National Health and Nutrition Examination Survey (NHANES) (wave 2021–2023) [17] and The Cancer Genome Atlas (TCGA) [18]. In brief, the NHANES study is a complex, stratified, multistage cluster probability sample of the civilian, noninstitutionalized population of the United States of America (USA). NHANES collects data through household interviews (for demographics, diet, tobacco use, and medical history) and a mobile examination centre (for health, dental, anthropometric, and biochemical examinations and biospecimen collection). For this study, we included the covariates age, gender, ethnicity, body mass index (BMI), total cholesterol levels, insulin levels, and self-reported myocardial infarction. We excluded participants who had missing data on these covariates, resulting in a dataset of 2,924 unique observations.

The Cancer Genome Atlas (TCGA) reflects the high-dimensional, heterogeneous data structures characteristic of multi-omics studies. TCGA is an ongoing collaboration between the National Cancer Institute and National Human Genome Research Institute in the USA with the aim of generating comprehensive, multi-dimensional maps of the key genomic changes in 33 types of cancer [18]. TCGA contains multi-omic data on tumour tissue and matched normal tissues from more than 11,000 patients alongside clinical information. For this study, we included data on tumour purity in bulk RNA sequencing (using the Illumina HiSeq 2000 platform), as determined from the ABSOLUTE algorithm [19] and a previously derived gene signature [20]. We included all tumour samples for which there were no missing values in gene expression, resulting in a dataset of 9,678 samples.

## Synthetic data generation and performance evaluation

We generated synthetic versions of both the NHANES and TCGA datasets using the following four data generators, selected to represent a range of generators both with and without specific privacy-preserving features, trained with their default hyperparameters: Private Aggregation of Teacher Ensembles Generative Adversarial Networks (PATE-GAN) [21], Denoising diffusion probabilistic models (DDPM) [22], Anonymization through Data Synthesis using Generative Adversarial Networks (ADS-GAN) [23], and Conditional Tabular Generative Adversarial Network (CTGAN) [24].

PATE-GAN, DDPM, and ADS-GAN are specifically privacy-preserving synthetic data generators. PATE-GAN uses differential privacy, a mathematical framework that ensures the inclusion or exclusion of a single data point does not significantly affect the output of data analysis [25]. We used a default privacy epsilon of 1 for these analyses based on previous work that has shown that this value can still provide strong privacy guarantees with minimal impact on the usefulness of the synthetic data [21]. As the epsilon is reduced, the privacy of the resulting synthetic data rises but with a trade-off in terms of its utility for downstream tasks such as prognostic modelling [21,26]. DDPM and ADS-GAN are specifically tailored to protect against re-identification attacks, where an adversary attempts to identify individuals within anonymised or pseudonymised datasets. CTGAN has no additional privacy-preserving framework embedded. Many generators will produce synthetic data that preserve privacy. However, generators that use differential privacy or protect against reidentification attacks provide users with specific, tuneable, mathematical guarantees over the privacy and fidelity of the resulting synthetic data. Such guarantees do not, on their own, reduce the need for rigorous evaluation of the privacy of the datasets generated.

To evaluate the quality, utility, and privacy of the generated data, we focused on several performance metrics (an exhaustive list of metrics and results are presented in Table B in S1 Text) [9,27,28]. For data quality, we used several metrics including Jensen-Shannon distance, the empirical maximum mean discrepancy, and α-precision [27]. To evaluate the utility of the data for downstream tasks, we then built predictive models and measured the mean squared error (MSE) (for regression) and the area under the received operating characteristic (AUC) curve (for classification) on both training and out-of-distribution data. For data privacy, we measured its authenticity, k-anonymity, and identifiability score. Authenticity quantifies the percentage of generated samples that are not near-identical copies of any real training example [27].

K-anonymization involves quantifying the smallest k such that, in the real data (for ground truth) or the synthetic data, every combination of quasi-identifiers appears at least k times. The identifiability score assesses how easily a synthetic record can be linked back to a specific real individual by comparing quasi-identifiers between real and synthetic datasets to test for privacy in the synthetic dataset.

### Synthetic data augmentation

In common with many research cohorts, NHANES does not have equal representation across different ethnicities. To address this, we used a four-stage augmentation pipeline: baseline enumeration; enrichment target calculation; conditional synthetic generation; and cohort integration and evaluation. This allowed us to train models on augmented datasets containing the same number of cases from each ethnic group represented. Further details can be found in the Supplementary Methods.

### Statistical analysis

To compare distributions of variables between real and synthetic datasets, we calculated the counts and percentages (for categorical variables) and medians and interquartile ranges (IQR) (for continuous variables) in the real and synthetic datasets in the NHANES dataset and mean and standard deviation for the gene expression in the TCGA dataset.

For the real and synthetic NHANES databases, we modelled the relationship between self-reported myocardial infarction using logistic regression and including all other variables in the dataset as covariables. We obtained odds ratios (OR) and 95% confidence intervals (CI) from these models. We used ethnicity categorisations as provided by NHANES. We used a complete case analysis. We subsequently built prediction models, analysing discriminative performance (area under the receiver operating curve; AUC) in aggregate and across sub-groups using bootstrapped (1,000 runs) confidence intervals. We present results from models that were trained on synthetic data and tested on the real dataset. For the TCGA dataset, we calculated the MSE of the predicted purity. We used a 5-fold cross-validation with 80% training and 20% testing data in each fold reporting the mean error across the folds alongside the standard deviation.

## Results

### Comparison of SynthCraft against standalone LLMs

SynthCraft used chain-of-thought reasoning to adapt its workflow to generate high-quality synthetic versions of both a tabular epidemiological dataset – NHANES – and a genomic dataset – TCGA (Fig 1). A complete example of this workflow can be found in Fig A in S1 Text.

To assess the value of SynthCraft over the use of state-of-the-art LLMs (GPT-5), we performed three ablation studies with the NHANES dataset. In each study, we used an instance of GPT-5 equipped with access to the same tools as SynthCraft so that we could isolate the advantages of the reasoning engine in orchestrating an end-to-end research workflow. In none of the ablation studies was GPT-5 alone able to complete the analyses (Table 1). Different error types occurred in each study, ranging from ignoring the Synthcity tool and trying to write its own code to generate synthetic data (i.e., bypassing established methods), through attempting to generate synthetic data using all possible generators available in the Synthcity package and overwhelming the compute available, to failing to evaluate the synthetic data. Importantly, in none of the ablation studies did GPT-5 involve the human user at critical stages in the workflow, which could lead to errors and inappropriate analyses.

### Results of synthetic data generated from NHANES using SynthCraft

We first compared the quality of the different datasets generated to the original NHANES dataset. The synthetic datasets generated with ADS-GAN, CTGAN, and DDPM, had similar statistical measures of fidelity to each other (Table C in S1 Text), whilst replicating the variable distributions seen in the original dataset (Table 2). Despite consistency at an

**Table 1. Ablation studies comparing SynthCraft against GPT-5.**

| Workflow action | SynthCraft | Ablation studies | | |
| --- | --- | --- | --- | --- |
| | | GPT-5 run 1 | GPT-5 run 2 | GPT-5 run 3 |
| Upload data file | ✔ | ✔ | ✔ | ✔ |
| Exclude/keep columns | ✔ | X | X | X |
| Perform exploratory data analysis | ✔ | ✔ | ✔ | ✔ |
| Generate descriptive statistics | ✔ | ✔ | ✔ | ✔ |
| Confirm ML problem type | ✔ | ✔ – This is the only point GPT-5 allowed the user to do anything. | ✔ | ✔ |
| Offer the user privacy plugins | ✔ | ✔ | X – GPT-5 misunderstood and tried to write its own code. Gave user confusing instructions | X |
| Select synthetic data generation method | ✔ | X – Ran all Synthcity generator methods without checking with user what they want | – | ✔ |
| Offer augmentation | ✔ | ✔ | – | X |
| Generate synthetic data | ✔ | X – Errored during generation by overloading the system trying to use all methods in Synthcity | – | ✔ |
| Calculate the metrics for the generated data | ✔ | – | – | ✔ |
| Explain and compare metrics for different methods | ✔ | – | – | X |
| Perform exploratory data analysis | ✔ | – | – | X |
| Generate descriptive statistics | ✔ | – | – | X |
| Plot synthetic data | ✔ | – | – | ✔ |
| Summarise the project | ✔ | – | – | X – Also tries and fails to run its own evaluation pipeline. |
| **Other issues** | | | | |
| Important steps require confirmation from user | ✔ | X | X | X |
| Calls correct tools in response to feedback | ✔ | ✔ | X | ✔ |
| No useless tool calls | ✔ | ✔ | ✔ | ✔ |
| No code generated that fails to run | ✔ | ✔ | X | X |
| Explains the different generators | ✔ | X | X | X |
| Quality synthetic data comparison | ✔ | X | X | X – fails to run own evaluation |
| Detailed description of the methods to choose between | ✔ | X | X | X |

Ablation studies comparing the performance of SynthCraft by comparison with the use of GPT-5 (OpenAI) equipped with the same tools. This provides an evaluation of the value added by SynthCraft's reasoning engine.

aggregate level, all three models showed some departure from the original dataset in the proportion of individuals with the outcome of interest - myocardial infarction - with 3.1%, 2.6%, and 7.4% of the synthetic cohorts generated using ADS-GAN, CTGAN, and DDPM, respectively, having the outcome relative to 4.2% of the real cohort. By contrast, the data generated by PATE-GAN had both lower statistical measures of fidelity and more divergence from the descriptive characteristics of the NHANES data, but would be considered more private, with greater k-anonymity and authenticity metrics, and the lowest identifiability scores (Table C in S1 Text).

We then modelled the relationship between the covariates and the occurrence of myocardial infarction in each dataset using logistic regression (Table 3). In keeping with the statistical measures of fidelity and descriptive statistics, regression models generated using synthetic data broadly preserved the relationships between variables and outcome seen in the real dataset. ADS-GAN and CTGAN reproduced the parameter estimates found in the real dataset most closely, however there were notable discrepancies in the relationship between ethnic group and myocardial infarction.

We subsequently analysed the discriminative performance of logistic regression models trained on synthetic data when tested on real data (Fig 2). Models trained on purely synthetic data generated using ADS-GAN and CTGAN showed near equivalent AUCs to a model trained on the original dataset (real: 0.818, 95% confidence intervals [CI]: 0.773-0.859; ADS-GAN: 0.781, 95% CI: 0.733-0.827; CTGAN: 0.797 95% CI: 0.746-0.847), although models trained on PATE-GAN and DDPM performed less well (Table D in S1 Text). These patterns were maintained when analysing performance by age and ethnicity sub-groups (Fig 2 and Table D in S1 Text).

## The impact of data augmentation on model performance

Augmentation – where synthetic data are added to the original data to reduce imbalances in particular features – is an important use-case for synthetic data. We thus asked SynthCraft to undertake ethnicity-specific augmentation of the

**Table 2. Comparison of variable distributions in the real and synthetically generated datasets (NHANES dataset).**

|  | Real dataset | Synthetic datasets | | | |
|  |  | ADS-GAN | CTGAN | DDPM | PATE-GAN |
| --- | --- | --- | --- | --- | --- |
| Age | 58 (40-68) | 57 (33-63) | 52 (33-66) | 57 (40-69) | 45 (43-54) |
| Gender |  |  |  |  |  |
| Male | 1311 (44.87) | 1312 (44.90) | 1269 (43.43) | 1200 (41.07) | 1477 (50.55) |
| Female | 1611 (55.13) | 1610 (55.10) | 1653 (56.57) | 1722 (58.93) | 1445 (49.45) |
| Ethnicity |  |  |  |  |  |
| Mexican American | 209 (7.15) | 212 (7.26) | 214 (7.32) | 200 (6.84) | 65 (2.22) |
| Other Hispanic | 315 (10.78) | 332 (11.36) | 410 (14.03) | 289 (9.89) | 90 (3.08) |
| Non-Hispanic White | 1769 (60.54) | 1817 (62.18) | 1688 (57.77) | 1735 (59.38) | 2573 (88.06) |
| Non-Hispanic Black | 305 (10.44) | 266 (9.10) | 289 (9.89) | 424 (14.51) | 92 (3.15) |
| Other Ethnicity | 324 (11.09) | 295 (10.10) | 321 (10.99) | 274 (9.38) | 102 (3.49) |
| Body Mass Index (kg/m$^2$) | 28.40 (24.60-33.30) | 28.79 (24.53-33.65) | 27.93 (24.28-33.13) | 27.63 (23.72-32.80) | 19.82 (16.46-23.45) |
| Total Cholesterol (mmol/L) | 4.81 (4.09-5.56) | 5.19 (4.36-5.83) | 5.42 (4.57-5.93) | 4.66 (3.87-5.45) | 6.19 (5.34-6.89) |
| Insulin (µU/mL) | 9.28 (5.86-15.52) | 11.43 (8.44-21.73) | 6.34 (4.56-13.61) | 8.84 (4.96-15.30) | 18.65 (18.65-28.29) |
| Self-reported myocardial infarction | 122 (4.18) | 90 (3.08) | 77 (2.64) | 216 (7.39) | 66 (2.26) |

All data are either *n* (%) for categorical variables or median (interquartile range) for continuous variables. The distribution of variables in the original selection of data from the NHANES (i.e., the real dataset) and four synthetic datasets from various methods are given. All synthetic datasets were generated using SynthCraft.

Abbreviations: ADS-GAN, Anonymization through Data Synthesis using Generative Adversarial Networks; CTGAN, conditional table generative adversarial network; DDPM, denoising diffusion probabilistic models; NHANES, National Health and Nutrition Examination Survey; PATE-GAN, Private Aggregation of Teacher Ensembles Generative Adversarial Networks.

**Table 3. Comparison of regression parameters for self-reported myocardial infarction estimates in the real and synthetically generated datasets (NHANES dataset).**

| | Real dataset | Synthetic datasets | | | |
| --- | --- | --- | --- | --- | --- |
| | | ADS-GAN | CTGAN | DDPM | PATE-GAN |
| Age (per year) | 1.06 (1.05-1.08) | 1.04 (1.02-1.05) | 1.04 (1.02-1.05) | 1.12 (1.10-1.14) | 1.02 (1.00-1.04) |
| Gender | | | | | |
| Male | Ref | Ref | Ref | Ref | Ref |
| Female | 0.58 (0.39-0.87) | 0.51 (0.32-0.80) | 0.75 (0.47-1.20) | 0.85 (0.57-1.26) | 1.12 (0.69-1.84) |
| Ethnicity | | | | | |
| Mexican American | Ref | Ref | Ref | Ref | Ref |
| Other Hispanic | 2.46 (0.78-7.75) | 0.72 (0.27-1.91) | 0.62 (0.20-1.99) | 1.80 (0.48-6.73) | 1.53 (0.36-6.39) |
| Non-Hispanic White | 1.45 (0.51-4.16) | 0.58 (0.27-1.22) | 0.86 (0.35-2.09) | 0.95 (0.28-3.23) | 0.42 (0.13-1.40) |
| Non-Hispanic Black | 1.59 (0.49-5.14) | 0.56 (0.21-1.51) | 1.37 (0.51-3.70) | 5.27 (1.50-18.46) | 0.71 (0.14-3.68) |
| Other Ethnicity | 1.47 (0.44-4.88) | 0.30 (0.09-1.03) | 1.15 (0.40-3.29) | 0.85 (0.21-3.49) | 1.41 (0.35-5.72) |
| Body Mass Index (per kg/m$^2$) | 1.02 (0.99-1.05) | 1.06 (1.03-1.10) | 1.01 (0.98-1.05) | 0.89 (0.86-0.91) | 1.05 (1.00-1.09) |
| Total Cholesterol (per mmol/L) | 0.53 (0.43-0.65) | 0.52 (0.40-0.67) | 0.45 (0.33-0.61) | 0.77 (0.69-0.86) | 1.14 (0.94-1.38) |
| Insulin (per μU/mL) | 1.00 (1.00-1.01) | 0.99 (0.98-1.00) | 1.00 (0.98-1.01) | 1.01 (1.00-1.01) | 1.00 (0.99-1.00) |

All data are either *n* (%) for categorical variables or median (interquartile range) for continuous variables. The distribution of variables in the original selection of data from the NHANES (i.e., the real dataset) and four synthetic datasets from various methods are given. All synthetic datasets were generated using SynthCraft.

Abbreviations: ADS-GAN, Anonymization through Data Synthesis using Generative Adversarial Networks; CTGAN, conditional table generative adversarial network; DDPM, denoising diffusion probabilistic models; NHANES, National Health and Nutrition Examination Survey; PATE-GAN, Private Aggregation of Teacher Ensembles Generative Adversarial Networks; ref, reference group.

original NHANES data. However, training predictive models of self-reported myocardial infarction on these augmented datasets did not improve either overall or sex- or ethnicity-specific discriminative performance (Fig 2 and Tables D-E in S1 Text). Indeed, augmentation with synthetic data across all synthetic data generators led to reductions in the overall AUC and discriminative performance by subgroup by comparison with using the real data alone. Because the performance of models built on the original data were both high (AUC > 0.8) and consistent across subgroups, these findings are not unexpected. They do highlight the need for iterative testing of synthetic data both across generators and use cases, and that synthetic data can balance representation across groups, but that does not necessarily remove bias or fairness [29].

## Results of synthetic data generated from TCGA using SynthCraft

Our second use case for SynthCraft was in genomic data. Models trained on synthetic genomic data predict tumour purity. For the TCGA cohort, we generated five synthetic versions using PATE-GAN, ADS-GAN, CTGAN, and DDPM via the SynthCraft platform. Quality metrics for each synthetic dataset are summarized in Table F in S1 Text. We then compared the distributions of gene expression levels and tumour purity estimates between the real and synthetic cohorts, finding close alignment across all methods. To assess downstream analytical fidelity, we applied a previously identified gene signature [20] in an XGBoost regression model to predict tumour purity. Predictive performance was comparable between the real data and all synthetic datasets except for PATE-GAN, which underperformed.

We repeated the same synthetic data generation pipeline for the TCGA cohort, demonstrating that SynthCraft can equally process high-dimensional genomic data. We found similar patterns to those in NHANES, where the statistical fidelity and utility of the synthetic data varied by generator (Tables G-H in S1 Text). XGBoost regression models predicting tumour purity had comparable mean squared errors when trained on synthetic data and the original TCGA cohort, except for models trained on synthetic data generated by PATE-GAN (Fig 3 and Table H in S1 Text).

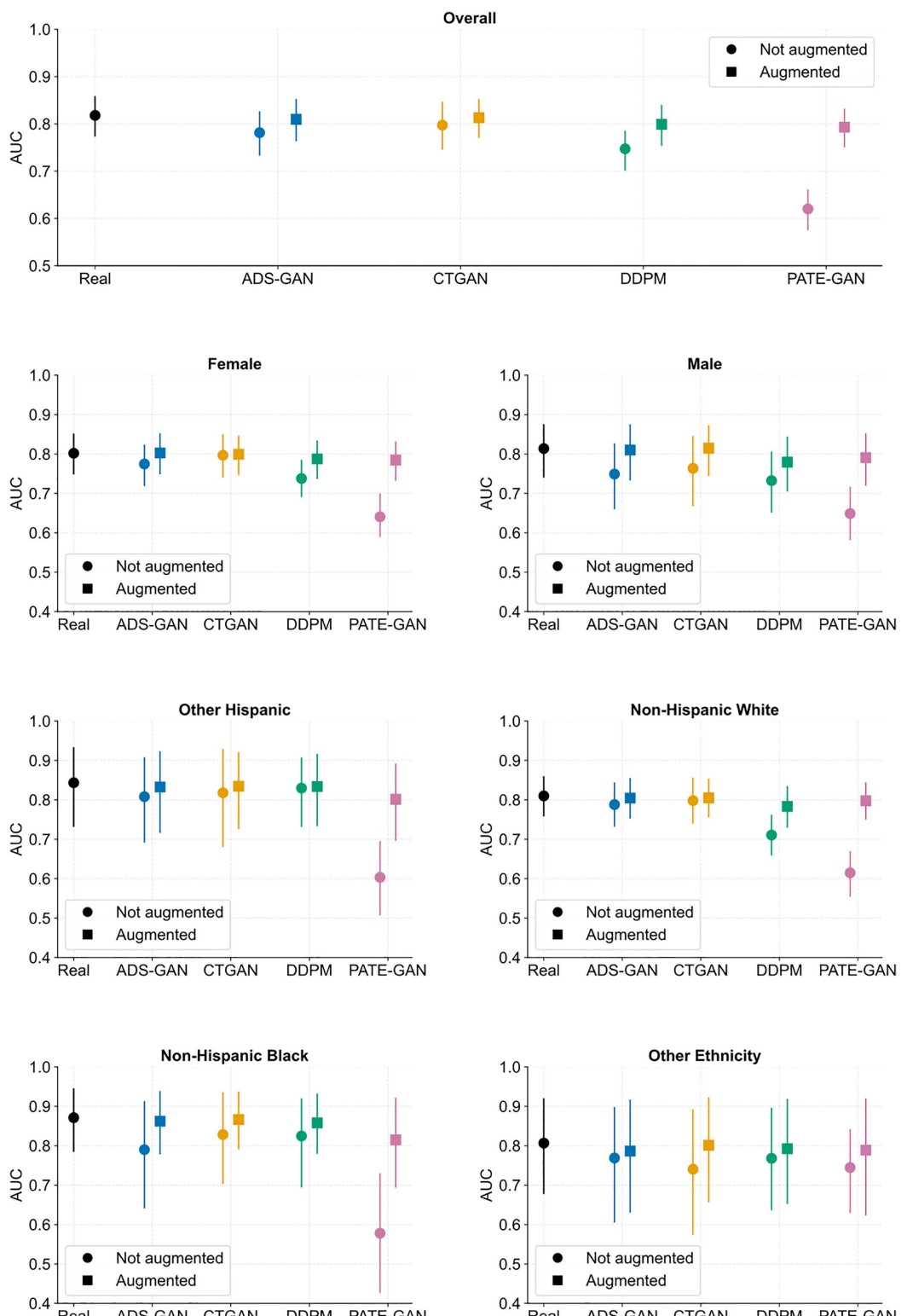

**Fig 2. Discriminative performance (AUC) overall and by sub-group of logistic regression models trained on the original (NHANES) dataset, purely synthetic datasets (circles), or original dataset with augmentation (squares) when tested on the original real dataset.** The discriminative performance of models trained purely on synthetic data can rival a model trained on real data, with the quality of models varying by synthetic dataset.

Augmentation of the real dataset did not improve performance relative to using the real dataset for model development without augmentation. Abbreviations: ADS-GAN, Anonymization through Data Synthesis using Generative Adversarial Networks; CTGAN, conditional table generative adversarial network; DDPM, denoising diffusion probabilistic models; NHANES, National Health and Nutrition Examination Survey; PATE-GAN, Private Aggregation of Teacher Ensembles Generative Adversarial Networks; AUC, area under the curve.

## Discussion

Despite its potential [2], the use of synthetic data in practice is complex, from the selection and training of the synthetic data generator to context-specific evaluation. We demonstrate that an AI-based partner can support the systematic use of state-of-the-art synthetic data generation methods to develop and evaluate synthetic data across both tabular and genomic datasets entirely through natural language.

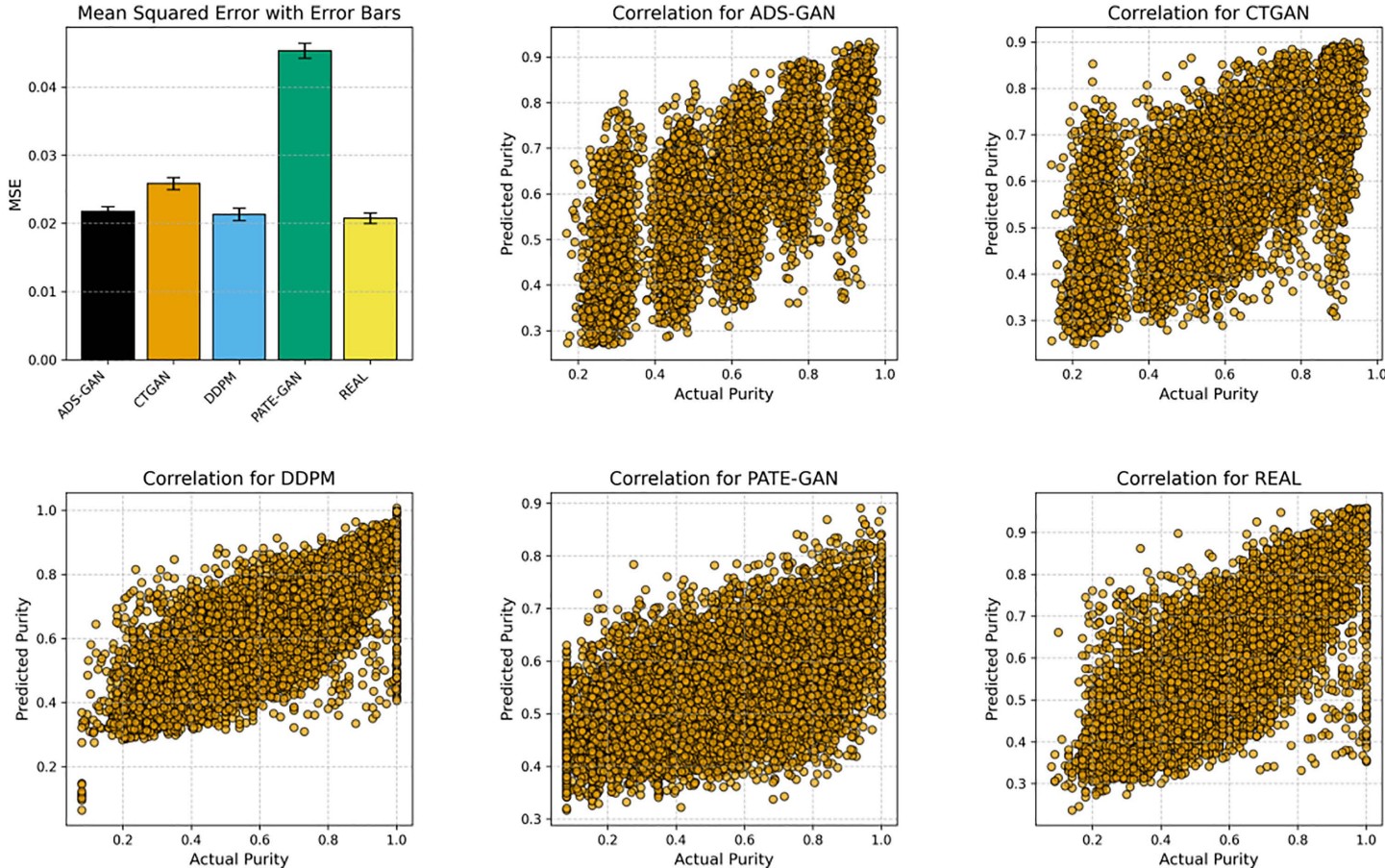

**Fig 3. Scatter plots illustrate the correlation between actual and predicted tumour purity for the real dataset and four synthetic datasets.** Correlations between gene expression levels and estimates of tumour purity were broadly similar across synthetic and real datasets, but the mean squared error of an XGBoost regression model predicting tumour purity was noticeably different when using synthetic data generated with PATE-GAN. Abbreviations: ADS-GAN, Anonymization through Data Synthesis using Generative Adversarial Networks; CTGAN, conditional table generative adversarial network; DDPM, denoising diffusion probabilistic models; NHANES, National Health and Nutrition Examination Survey; PATE-GAN, Private Aggregation of Teacher Ensembles Generative Adversarial Networks; MSE, mean squared error.

Improving the adoption, accessibility, reproducibility, and quality of research methods has relied upon improved training and a proliferation of research checklists and guidelines. These approaches are inherently limited: although guidelines can provide an approach to tackling a problem, there are few mechanisms to ensure their systematic use either by researchers or publishers [14]. Furthermore, it has been suggested that nearly 95% of time required to conduct analyses involving machine learning is spent programming, a technical debt [30] that slows the dissemination of new methodological advances or potentially restricts access to teams with sufficient resources with contributing necessarily to scientific advance.

We show here that LLMs, specifically the use of agent-based frameworks, could support an alternative approach [31] in which researchers interact with AI partners that encode and standardise good practice, ensuring that recommendations from guidelines and research checklists are considered, improving the quality of research performed. This approach simultaneously improves the accessibility of state-of-the-art ML methods: the user is no longer required to be an expert programmer, to be limited to synthetic data generators with which they have familiarity, or indeed to only those that are available in a single programming language.

Agent-based frameworks build on the potential for LLMs to act as reasoning engines [32], with guardrails enforced through access only to pre-specified methodological software packages. This ensure that the resulting analyses are technically sound and is supported by the transparent reporting of any code run. A crucial feature of this approach is collaboration between the user and the framework – human-in-the-loop iteration – providing SynthCraft with domain knowledge to contextualise the problem; SynthCraft is a partner that augments a researcher, not a replacement.

Both SynthCraft and the underlying Synthcity package used to generate the synthetic data itself are open source, such that they verified, improved, or even adapted by the broader research community. As new synthetic data methods become available, these new tools can be incorporated into SynthCraft without requiring re-training of the underlying reasoning framework. We have demonstrated the creation of synthetic data for data access and augmentation, but SynthCraft is not limited to these, with any use case supported by the underlying Synthcity package. As with the underlying LLMs, Synthcity itself could be swapped for an alternative synthetic data generating package. SynthCraft has a specific purpose: the generation of synthetic data. This could be extended by linking other agent-based frameworks, for example to create downstream prediction models, which could operate in a cooperative manner to create end-to-end analytic pipelines.

SynthCraft has some limitations. We found that the framework can require prompting to continue to the next stage of analyses, becoming focussed on the specific task at hand. With future research into training agent-based workflows and the ongoing improvement in underlying LLMs, we expect this limitation to be less prevalent in the future. Although the underlying LLM does not have access inherently to the data, this safeguard could be bypassed by user prompting. Controlled instances of LLMs are available from cloud services that are compliant with healthcare privacy regulations and should be used to add security to the system; we provide instructions on how to set up SynthCraft with this feature built-in. SynthCraft is also capable of being used with open-weight or open-source LLMs, providing choice over how the system is deployed and allowing for more granular privacy controls. In the development of SynthCraft we discussed the prototype with individuals from different professional backgrounds and levels of programming proficiency but have not conducted a formal user study. This will be the subject of future work. Furthermore, as an open-source project, we encourage users to suggest improvements and contribute to the development of the software. In our analyses, we demonstrate the system with two different datasets (epidemiological and genetic). Future work could test how the system performs across a broader range of datasets, including of different scales.

In conclusion, we present an AI partner – SynthCraft – that enables the generation of synthetic datasets and augmented synthetic datasets using natural language. Such AI partners hold promise in democratising access to state-of-the-art methodologies, whilst improving the quality and reproducibility of analyses, furthering a new approach to scientific analysis.

## Supporting information

**S1 Text. Supporting Information.** Table A. Summary of synthetic data generator tools available in SynthCraft. Table B. Data quality metrics in SynthCraft. Table C. Performance metrics for generated synthetic datasets (NHANES). Table D. Discrimination (AUC) by sub-group for logistic regression models trained on the real data and on synthetic data (NHANES dataset). Table E. Discrimination (AUC) by sub-group for logistic regression models trained on the real data and on real data augmented with synthetic data (NHANES dataset). Table F. Exhaustive list of performance metrics for generated synthetic datasets (TCGA dataset). Table G. Comparison of variable distributions in the real and synthetically generated datasets (TCGA purity dataset). Table H. Performance on the TCGA gene purity dataset. Table I. Ablation studies comparing SynthCraft against GPT-4o. Fig A. Workflow for the NHANES dataset.
(PDF)

## Author contributions

**Conceptualization:** Thomas Callender, Anders Boyd, Mihaela van der Schaar.

**Data curation:** Robert Davis, Silas Ruhrberg Estevez.

**Formal analysis:** Thomas Callender, Anders Boyd, Robert Davis, Silas Ruhrberg Estevez.

**Funding acquisition:** Mihaela van der Schaar.

**Investigation:** Thomas Callender, Anders Boyd, Mihaela van der Schaar.

**Methodology:** Thomas Callender, Anders Boyd, Robert Davis, Silas Ruhrberg Estevez, Mihaela van der Schaar.

**Project administration:** Thomas Callender, Mihaela van der Schaar.

**Software:** Robert Davis.

**Supervision:** Thomas Callender, Mihaela van der Schaar.

**Validation:** Anders Boyd, Robert Davis.

**Visualization:** Thomas Callender, Robert Davis, Silas Ruhrberg Estevez.

**Writing – original draft:** Thomas Callender, Anders Boyd, Robert Davis, Silas Ruhrberg Estevez.

**Writing – review & editing:** Thomas Callender, Anders Boyd, Robert Davis, Silas Ruhrberg Estevez, Juan M. Lavista Ferres, Mihaela van der Schaar.

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
