## [Decision Letter · Decision Letter 0]

2 Dec 2025

Response to Reviewers
Revised Manuscript with Track Changes
Manuscript
**Journal Requirements:**

1. Please provide an Author Summary. This should appear in your manuscript between the Abstract (if applicable) and the Introduction, and should be 150–200 words long. The aim should be to make your findings accessible to a wide audience that includes both scientists and non-scientists. Sample summaries can be found on our website under Submission Guidelines: [LINK]

https://journals.plos.org/digitalhealth/s/submission-guidelines#loc-parts-of-a-submission

**Additional Editor Comments (if provided):**
**Reviewers' Comments:**

**Comments to the Author**

1. Does this manuscript meet PLOS Digital Health’s publication criteria?

Reviewer #1: Yes

Reviewer #2: Yes

Reviewer #3: Yes

2. Has the statistical analysis been performed appropriately and rigorously?

Reviewer #1: Yes

Reviewer #2: Yes

Reviewer #3: Yes

3. Have the authors made all data underlying the findings in their manuscript fully available (please refer to the Data Availability Statement at the start of the manuscript PDF file)?

Reviewer #1: Yes

Reviewer #2: Yes

Reviewer #3: Yes

4. Is the manuscript presented in an intelligible fashion and written in standard English?

Reviewer #1: Yes

Reviewer #2: Yes

Reviewer #3: Yes

Reviewer #1: This is a great and timely contribution to the field of digital health and AI for biomedical data science. The authors introduce SynthCraft, a human-in-the-loop AI partner that leverages LLM reasoning and reinforcement learning to orchestrate synthetic data generation, evaluation, and reporting. The system lowers technical barriers for researchers, promotes transparency, and enhances reproducibility in data access and augmentation workflows.

Suggestions for improvement:

-Clarify LLM involvement and reproducibility:

-Specify the reproducibility of results given that GPT-4 (temperature = 0.5) introduces nondeterminism. Would different LLM versions or parameters affect workflow outcomes?

-Explicitly state how prompts and agent reasoning are logged or shared for transparency.

-Quantify computational efficiency and scalability:

-Include run times or resource requirements for SynthCraft workflows on both datasets to guide prospective users.

-Expand on limitations of privacy evaluation: While metrics like k-anonymity and authenticity are discussed, consider including a brief analysis or discussion of differential privacy guarantees across the models.

-Discuss user experience and usability testing: Since SynthCraft aims to democratize access to synthetic data generation, some preliminary feedback from non-expert users (clinicians, data scientists) would strengthen the claim of accessibility.

Reviewer #2: - In Table 1, the authors have reported the units of Insulin as uU/mL. Do you mean micro-U/mL? Please use μ instead of u to avoid any confusion. the same problem is present in Table 2.

- In the methods, dataset section authors have declared to include waist circumference from NHANES data set but the data was not reported in results section.

- The major innovation of this study is the integration of a LLM model with four types of generator models. The idea is simple yet effective, but the problem with LLM models is the incorrect suggestions from the generator model. the LLM may increase the bias and inaccuracy by hallucination. there is also the problem of keeping the LLM model updated. To these reasons introduction of LLM model may increase the needed computational power, increased inaccuracy, and reduce tractability in the generated data.

Reviewer #3: Clarify the core contribution

The manuscript should more clearly articulate what is methodologically novel about SynthCraft beyond integrating existing tools (Synthcity + LLMs). A concise bullet list of contributions at the end of the Introduction would help distinguish between (a) the agentic framework and (b) the empirical comparison of synthetic data generators.

Detail the agent / bandit reasoning module

The description of the episodic multi-armed bandit is too high-level. Please formalize the state, action space, reward function, and learning procedure, and explain how this component actually influences pipeline decisions compared with simple rule-based orchestration.

Evaluate SynthCraft as a system (not only generators)

Most results compare CT-GAN, ADS-GAN, DDPM, and PATE-GAN, but do not quantify the benefit of SynthCraft itself. Please add an evaluation comparing SynthCraft to (i) manual use of Synthcity or (ii) a simple non-bandit LLM agent, in terms of time, coverage of analyses, or error rate.

Add an ablation or user-in-the-loop study

To support the “AI partner” claim, a small user study or ablation (e.g., with/without bandit, with/without agents) would be very valuable. Even a pilot study with a few analysts showing reduced effort or improved completeness of analyses would strengthen the practical impact.

Improve methodological transparency and reproducibility

Please provide more implementation details: exact Synthcity and library versions, LLM model name, temperature, max tokens, prompt templates, and number of tool calls per episode. Consider placing full prompts and configuration files in the Supplementary Material or a public repository.

Clarify dataset preprocessing and splits

For both NHANES and TCGA, the preprocessing steps (feature selection, normalization, handling of missing data, train/test splits) should be described more precisely. This includes how outcome variables are defined, how ethnic groups or tumour purity labels are handled, and whether any information leakage might occur.

Deepen the analysis of privacy–utility trade-offs

The comparison of PATE-GAN with non-DP generators is interesting but somewhat superficial. Please expand the discussion on how ε was chosen, what ε≈1 implies in terms of privacy strength, and how utility degrades as privacy is strengthened. A small sensitivity analysis over ε would be very informative if feasible.

Expand fairness and subgroup analyses in NHANES

The ethnicity-based augmentation experiment is promising, but the evaluation currently focuses mainly on overall AUC. Please report subgroup metrics (e.g., AUC or calibration per ethnic group, equal opportunity gaps) to show whether augmentation helps or harms fairness across groups.

Clarify LLM data exposure and deployment constraints

The manuscript claims that the LLM “never sees the raw data,” but this is not fully demonstrated. Please include concrete examples of prompts, explain how data are abstracted into code/text safely, and discuss safeguards to prevent users from accidentally pasting identifiable data into the LLM interface (especially in clinical deployments).

Strengthen the limitations and future work section

The limitations section could be more explicit. For example: reliance on two datasets, absence of a formal usability study, dependence on proprietary LLMs, and the lack of automated hallucination detection or safety filters. Also consider outlining concrete future directions (e.g., open-weight models, more domains, better safety checks).

Improve figure and table clarity

Some figures and tables would benefit from clearer legends and more self-contained captions. Please ensure that all axes, groups, and generators are clearly labeled, use marker shapes or line styles that are robust to colour-blindness, and state in each caption what the main takeaway is.

Language, style, and consistency edits

There are minor grammatical issues and inconsistent terminology (e.g., variations in the naming of generators such as PATE-GAN / PATEGAN). A careful language pass to fix grammar, streamline long sentences, and standardize terminology will improve readability and professionalism.

**Do you want your identity to be public for this peer review?** For information about this choice, including consent withdrawal, please see our Privacy Policy

Reviewer #1: No

Reviewer #2: **Yes:** Hamidreza Ashayeri

Reviewer #3: No

**Figure resubmission:**

**Reproducibility:** To enhance the reproducibility of your results, we recommend that authors of applicable studies deposit laboratory protocols in protocols.io, where a protocol can be assigned its own identifier (DOI) such that it can be cited independently in the future. Additionally, PLOS ONE offers an option to publish peer-reviewed clinical study protocols. Read more information on sharing protocols at https://plos.org/protocols?utm_medium=editorial-email&utm_source=authorletters&utm_campaign=protocols

---

## [Decision Letter · Decision Letter 1]

21 Feb 2026

SynthCraft: an AI partner for synthetic data generation to support data access and augmentation in healthcare

PDIG-D-25-00704R1

Dear Dr Callender,

We are pleased to inform you that your manuscript 'SynthCraft: an AI partner for synthetic data generation to support data access and augmentation in healthcare' has been provisionally accepted for publication in PLOS Digital Health.

Best regards,

Hanieh Razzaghi

Section Editor

PLOS Digital Health

**Additional Editor Comments (if provided):**

**Reviewer Comments (if any, and for reference):**

Reviewer's Responses to Questions

**Comments to the Author**

Reviewer #2: All comments have been addressed

Reviewer #3: All comments have been addressed

publication criteria?

Reviewer #2: Yes

Reviewer #3: (No Response)

3. Has the statistical analysis been performed appropriately and rigorously?

Reviewer #2: Yes

Reviewer #3: (No Response)

4. Have the authors made all data underlying the findings in their manuscript fully available (please refer to the Data Availability Statement at the start of the manuscript PDF file)?

Reviewer #2: Yes

Reviewer #3: (No Response)

5. Is the manuscript presented in an intelligible fashion and written in standard English?

Reviewer #2: Yes

Reviewer #3: (No Response)

Reviewer #2: I appreciate the authors efforts for this manuscript. It was good at first submission and after all these revisions it has become an even better research item. All comments have been addressed and SynthCraft can prove to be useful tool in healthcare.

Reviewer #3: (No Response)

**Do you want your identity to be public for this peer review?** For information about this choice, including consent withdrawal, please see our Privacy Policy

Reviewer #2: **Yes:** Hamidreza Ashayeri

Reviewer #3: None
